# Effect of Microwave and Conventional Modes of Heating on Sintering Behavior, Microstructural Evolution and Mechanical Properties of Al-Cu-Mn Alloys

**DOI:** 10.3390/molecules26123675

**Published:** 2021-06-16

**Authors:** A. Muthuchamy, Muthe Srikanth, Dinesh K. Agrawal, A. Raja Annamalai

**Affiliations:** 1Department of Metallurgical and Materials Engineering, NIT, Tiruchirapalli 620015, Tamil Nadu, India; muthuchamy@nitt.edu; 2Centre for Innovative Manufacturing Research, VIT, Vellore 632014, Tamil Nadu, India; muthe.srikanth@vit.ac.in; 3Materials Research Institute, The Pennsylvania State University, State College, PA 16802, USA

**Keywords:** aluminum alloys, conventional sintering, microwave sintering, mechanical properties, corrosion resistance

## Abstract

In this research, we intended to examine the effect of heating mode on the densification, microstructure, mechanical properties, and corrosion resistance of sintered aluminum alloys. The compacts were sintered in conventional (radiation-heated) and microwave (2.45 GHz, multimode) sintering furnaces followed by aging. Detailed analysis of the final sintered aluminum alloys was done using optical and scanning electron microscopes. The observations revealed that the microwave sintered sample has a relatively finer microstructure compared to its conventionally sintered counterparts. The experimental results also show that microwave sintered alloy has the best mechanical properties over conventionally sintered compacts. Similarly, the microwave sintered samples showed better corrosion resistance than conventionally sintered ones.

## 1. Introduction

A wide range of applications is provided and its alloys, with a unique combined advantage which makes it the material to choose for many applications such as aerospace, the automotive, military, etc. due to their low density, coefficient of thermal expansion, high strength, wear-resistance, and improved damping properties [1,2]. Aluminum–metal matrix composites have received comprehensive attention for functional and fundamental reasons among the materials of tribological significance. Due to theirhigh compressibility, less density, specific strength, and economical processing, aluminum alloy composites can be quickly produced by the powder metallurgy technique compared with the other available fabrication techniques [3]. Aluminum alloys and aluminum-based metal matrix composites have found applications in the manufacture of various automotive engine components. The metal matrix composites’ main advantage is mechanical properties such as hardness and yield strength to be adequately regulated by strengthening the matrix and substantial mass control, which is necessary, mainly when used mutually [4]. Aluminum metal matrix composites are also used in the transportation sector due to their lower density, lower airborne emission, and less noise, which helps maintain the environmental regulations and provides good fuel economy [5]. The high strength to weight ratio of Aluminum matrix composites has also successfully cemented their place in military applications. The Young’s modulus of pure Aluminum can be enhanced by 300% (70 GPa to 240 GPa) by reinforcing the aluminum fibers [5]. Aluminum-based powder metallurgy alloys were used to produce near-net-shape products with high material utilization with less cost, lower processing temperature, and refined homogeneous microstructure with a lesser amount of porosity [6,7].

Further research into powered metal’s ability to absorb and dissipate microwave radiation has opened new powder metallurgy opportunities. Microwave sintering is an efficient, economic, and valuable approach to processing some P/M materials [8]. Microwave sintering has advantages like enhanced diffusion processes, reduced energy consumption, and rapid heating rates. Microwave sintering considerably reduces processing times, decreases sintering temperatures, improves physical and mechanical properties, and has lower environmental hazards. These are some features that have not been observed in conventional sintering [9,10].

Much research has been carried out for aluminum metal powder mixed with varied compositions of the alloying elements. Aluminum–manganese alloys and aluminum–coppers are two of the most-used combinations. Al-Mn alloys have elevated formability and corrosion resistance with high heat transfer coefficients, making them feasible for radiators, packaging, and roofing applications [11]. Such alloys have a very high strength-to-weight ratio and a density slightly higher than that of various plastics. Inter metallic-phase formation of Al-Mn elements by the adsorption of Mn within the liquid phase of Al acts as the driving force for solidification [12,13]. Al-Cu alloys are heat-treatable, and hence they possess high strength, especially at high homologous temperatures (200–300 °C), with higher toughness, resulting in a wide range of aircraft and transportation industry applications [14]. The compact density shows improvement after sintering due to precipitation due to the copper swaging nature of copper [15,16].

The present research attempts to study the effect of heating mode on the physical and mechanical properties of aluminum alloy composites produced through the powder metallurgy route. The effects of copper and manganese addition on the sinterability of the same were also studied.

## 2. Materials and Methods

For the present investigation, gas atomized pure elemental form aluminum, copper, and manganese powders were purchased from Krish Met Tech Pvt. Ltd.(Annamalai Colony St, Annamalai Colony, Virugambakkam, Chennai, Tamil Nadu, India) Chennai. By varying the Al, Cu, and Mn contents, three distinct alloys are designed, as shown in Table 1. The characteristics of the as-received powders are listed in Table 2. According to the designed composition, elemental powders are separately blended using a mortar for 60 min to obtaina uniform composition.

The powders were subjected to uniaxial compaction at 400 MPa and were made as cylindrical pellets (~25 mm diameter) using a Universal Testing Machine (model: Instron 8801, Norwood, MA, USA). Zinc stearate was used as a die wall lubricant [13]. All compacted alloys are sintered at 550 °C for 60 min using (i) conventional sintering in a tubular furnace (model: VBCC, TUBULAR FURNACE, Chennai, Tamilnadu, India) at a heating rate of 5 °C/min, and (ii) microwave sintering furnace (model: VBCC HYTERM FURNACE Chennai, Tamilnadu, India, multimode cavity 2.45 GHz, 6 kW) at a heating rate of 30 °C/min. Alumina boats were used for placing the samples inside the furnaces. Age-hardening of sintered samples was performed at 150 °C for 60 min in a box furnace (model: R257 INDFURN SUPERHEAT FURNACES, Chennai, Tamil Nadu, India)at a heating rate of 5 °C/min, and was water quenched to room temperature. By calculating the volume and weight of the pellets, the sintered densities of the pellets were determined. The densification parameter of the samples was calculated as:(1)$$\mathrm{Densification}\;\mathrm{parameter}=\frac{\mathrm{Sin}\mathrm{tered}\;\mathrm{density}-\mathrm{Green}\;\mathrm{density}}{\mathrm{Theoritical}\;\mathrm{density}-\mathrm{Green}\;\mathrm{density}}$$
(2)$$\mathrm{Radial}\;\mathrm{Shrinkage}=\frac{\mathrm{Green}\;\mathrm{diameter}-\mathrm{Sin}\mathrm{tered}\;\mathrm{diameter}}{\mathrm{Green}\;\mathrm{diameter}}$$
(3)$$\mathrm{Axial}\;\mathrm{Shrinkage}=\frac{\mathrm{Green}\;\mathrm{height}-\mathrm{Sin}\mathrm{tered}\;\mathrm{height}}{\mathrm{Green}\;\mathrm{height}}$$

The sintered samples were initially polished with 220, 400, 800, 1000, 1200, 1500, 1800, and 2000 grit SiCemery papers, progressively, and mirror polishing was done on the disc polishing machine with diluted alumina solutions on a velvet cloth. After polishing, the samples were etched using Keller’s reagent (3 mL-HNO_3_, 2 mL-HCl,1 mL-HF, and 100 mL-H_2_O) [17]. The polished sample microstructures were evaluated with optical microscopy (model: ZEISS-AXIO vert A1, Jena, Germany). Microhardness of all the compositions was tested using the Vickers hardness testing machine (model: MMT-X7B, no: MM5562X, Matsuzawa Co.,Ltd, Akita, Japan)with a load of 0.5 kgf, with 10 s dwell time. The indent’s diagonal lengths were measured, and the experiment was repeated three times to obtain a precise value.

The Vickers hardness of the sintered alloys is found out using the following equation:(4)$$\mathrm{Hv}=\frac{1.854\times P}{a^{2}}$$
where p = Load applied, a = Average length of the diagonal = $\frac{d_{1}{+d}_{2}}{2}$, d_1_ = length of diagonal 1, d_2_ = length of diagonal 2.

A scanning electron microscope [model: ZEISS-EVO18, Jena, Germany] was used in backscattered electron imaging mode for microstructural examination. The sample’s electrochemical activity was tested using the electrochemical method in a freely aerated 0.1N HCl solution. Before polarization, the polished samples were allowed to stabilize for 1 h to obtain stable open circuit potential (OCP). Electrochemical tests were carried out in a flat corrosion cell using a standard three-electrode configuration of the sample as the working electrode, platinum mesh as the counter electrode, and Ag/AgCl reference electrodes. Potentiodynamic polarization tests were carried out from −250 mV versus OCP to +1600 mV versus the reference electrode at a CT scan rate of 0.1667 mV. From the corresponding anodic and cathodic curves, the Tafel curves were created. The corrosion potential (E_corr_), corrosion current (I_corr_), and corrosion rate were determined from the polarization curves. The corrosion rate was determined by using the 1st-Stern method [18] and is expressed as follows:(5)$$\mathrm{Corrosion}\;\mathrm{rate}\;\left(\frac{\mathrm{mm}}{\mathrm{year}}\right)=0.0033\times \frac{e}{\rho }\times I_{\mathrm{corr}}$$
where e is the equivalent weight (g), ρ is the density within the material (mg/m^3^), and I_corr_ is the corrosion current (mA/m^2^).

## 3. Results & Discussion

### 3.1. Densification Response

The heating profiles of the compacts were as shown in Figure 1. The overall heating rate obtained in the microwave furnace was 30 °C/min at 550 °C for 60 min. concerning the heating cycle; a 34% reduction in the processing time was obtained during microwave sintering against the slower heating rate (5 °C/min) of a conventional furnace. A similar trend was depicted in the observations of C. Padmavathi et al. [16]. Table 3 shows the relative sintered density of the microwave and conventional sintered alloys. The microwave sintering technique yields better density as compared to a conventional sintering method. The presence of Aluminum enhances the density within the material. In contrast, Ananda Kumar et al. [4] validated that % porosity increased with a decrease in Al content for aluminum in the composites. After compaction, on heating the green compact just above the solidus temperature and high compressibility of aluminum, the particle starts to fuse, and hence densification takes place employing solid-phase sintering, which C. Padmavathi et al. explain [19]. During subsequent heating, semi-solid phases are formed at corresponding eutectic temperatures depending on the alloy composition. Higher sintering temperatures lead to the formation of a higher amount of the semi-solid phases, and hence further densification is possible, and the sample can be compressed to a greater extent. Similar behavior was observed for various aluminum powder compositions sintered at 550 °C to a higher sintered density (81.54% theoretical). Based on the weight percentage of aluminum in the composition, the rate of disintegration of particulate reinforcements followed by diffusion bonding and grain coarsening varies. The Al-57.28, Cu-18.6, and Mn-24.12 compositions have the lowest sinter-density among all the other compositions. The percentage of copper was directly proportional to the number of precipitates in the grains. A higher amount of densification was noted with the increase in copper content in both the sintered samples. At the time of aging, a further increase in precipitation was observed. The increase in the densification parameter was explained by DesalegnWogasoWolla et al. [15]. Weight and top layer loss in microwave sintering are comparatively less, as the heating was performed radially, outwards, within the sample. Since heat concentration was greater at the core, the sample’s surface was safe throughout the entire process, similar to what was explained by Morteza Oghbaei et al. [9].

### 3.2. Micro–Hardness

The variations in micro-hardness of samples, with varying Cu and Mn weight contents, sintered through conventional and microwave techniques are shown in Table 4. It was found that the addition of copper particulates could effectively enhance the micro-hardness as the elemental hardness of copper (369 HV) is higher than the other elements within the composition, and also because of the formation of precipitates due to aging, as described by J. Sun et al. [18]. In addition to this, Rainforth et al. [20] have also observed that the alloy A2124 (91.2Al-3.8Cu) exhibits more work hardening, while A3004 exhibited minimal work hardening. Beyond a specific percentage composition of copper, a decrease in hardness was observed due to a reduction in grain growth, limiting the semi-solid state’s flow within the aluminum grain boundaries. The results indicated that using the microwave sintering technique process had an overall advantage compared with the conventional sintering method considering the improvement in hardness, as reported by Morteza Oghbaei et al. [9]. From the SEM and optical microstructure of the Cu added alloys observed, copper is distributed equally in microwave sintered alloy. Still, in the conventional sintered alloys, copper is concentrated in some regions, and copper is not distributed uniformly throughout the alloy due to the high time given to the alloy during sintering. Even distribution of the Cu help to form a uniform distribution of precipitates, and these precipitates are influencing an increase in the hardness of the microwave sintered alloys.

### 3.3. Microstructural Results

The optical microstructures of conventionally and microwave sintered samples after polishing and etching are shown in Figure 2a–c. The microstructure revealed uniform distribution of Cu and Mn in the Al matrix. However, at some sites, clusters of Al-Cu phases were observed at the surface along with porosity. At 550 °C, aluminum melted and was distributed throughout the compact. This resulted in the formation of intermetallic phases along the grain boundaries. Dispersed Mn phases were found to be homogeneously distributed in all the samples of Al matrix made through the conventional sintering method, and the phenomenon is validated by Alexandra V. Khvan et al. [13]. The microwave processed sample possessed fewer amounts of pores, which are very small and uniformly distributed, while the conventionally sintered samples are found to have more porosity except in the case of Al-20%Cu-15%Mn (Figure 2c). The addition of Cu in Al-Mn alloy resulted in the formation of intermediate granules of Cu during the conventional mode of sintering, but it was found to be homogeneously dispersed between Al and Mn in the microwave due to lesser precipitation time, which was also observed by Morsi M. Mahmoud et al. [21]. Large intermediate granules were only formed during conventional sintering except in the case of Al-18.6%Cu-24.12%Mn composition. It was also found that the size of the Cu granules reduced in Al-20%Cu-15%Mn, as shown in Figure 2c. The availability of pure aluminum phases was lowest in the microwave sintered compact of Al-18.6%Cu-24.12%Mn composition, shown as the result of EDS (Table 5).

### 3.4. Scanning Electron Microscopy

Figure 3a–c shows the representative SEM micrograph of the sintered compacts in conventional and microwave mode. The micrograph shows that Mn and Cu particle sizes were small with microwave sintering as compared to conventional sintering. The SEM images in Figure 3a, representing Al-22.5%Mn composite, clearly revealed a uniform distribution of Mn particles within Al’s matrix in the case of microwave sintering, and it was similar to that reported by Morteza Oghbaei et al. [9]. Table 5 shows the EDS spectrum, which depicts Al, Cu, and Mn’s peaks and their respective elemental percentage. In all the compositions, the percentage of pure Mn was relatively more minor, which would account for the formation of various intermetallics between Al and Mn, as Alexandra V. Khvan et al. [13] described the solidification of Al-Mn intermetallics which led to the formation of Al_11_Mn_4_ and similar intermetallic phases, which was more likely during the consolidation process. Such intermetallic formation was observed profoundly in microwave sintering compared to conventional sintering and led to better physical and mechanical properties. M. Ellner et al. [22] described that MnAl_0.8_ showed a distinct behavior of brass-like ‘µ-phases,’ which was stable and expected to have a greater degree of mechanical properties. For both composites which contain copper, the less pure copper percentage, as depicted in EDS, resulted in less precipitation of copper at the grain boundaries in the case of conventional sintering as compared to microwave. LuboKloc et al. [21] described in their studies on aluminum alloy the high formation of inter-metallic of Al-Cu, which results in less availability of pure metals to precipitate and hence disturbs the mechanical properties of the sample. Smaller and more uniformly distributed intermetallic granules were observed in microwave sintering compared to the conventional mode of sintering, validated by Morsi M. Mahmoud et al. [13]. Tinier pore size and uniform distribution, which were observed in the SEM of microwave sintered sample, enabled us to confirm the possibility of higher and unified mechanical and physical properties, as expressed by MortezaOghbaei et al. [9].

### 3.5. Electrochemical Study

Table 6 represents the corrosion characteristics of the Al-Cu-Mn alloys obtained through conventional and Microwave sintering. Figure 4a–c represents the potential polarization curves of all the compositions sintered with conventional and microwave modes of sintering. As current density was inversely relative and directly proportional to the corrosion rate, it showed microwave sintering results yielded better corrosion resistance than conventional sintering in all study compositions. As the percentage porosity was reduced to microwave sintering, the corrosion rate was reduced due to the reduced anodic region on the sintered compact’s surface layers. The compact sintered in the microwave furnace stabilized at lower potentials, compared to be compact sintered in conventional. For conventional sintering, a higher corrosion potential (E_corr_), lower corrosion current density (I_corr_), and lower corrosion rate were observed, and a similar result was obtained by C. Padmavathi et al. [16]. Al-18.6%Cu-24.12% Mn composition showed a high corrosion rate when compared with remaining compositions. The result obtained from EDS analysis (Table 5) depicted that the availability of Cu and Mn in microwave sintering is more than conventional, indicating fewer chances of Al-Cu alloys Al-Mn alloy formation the grain boundaries, and it resulted in lesser corrosion rate. All liquid phase formation and alloy formation with Cu by adsorption phenomena led to a reduction of pure Al’s share on the surface level and sub-surface level in the case of conventional sintering. The open-circuit potentials of Al-Cu alloys, in solutions of near-neutral pH, were greater than or equal to the E_pit_ for pure Al over a broad range of Cu concentrations, as explained by Faith George et al. [23]. Deposition corrosion is a particular case of galvanic corrosion that takes the form of pitting. When particles of a more cathodic metal in the solution spread out on an aluminum surface to set up local galvanic cells, this confirms the widely held view that galvanic coupling of Al-Cu intermetallic particles with the Al promotes pitting and deposition corrosion.

Al-20%Cu-20-15%Mn composition shows higher corrosion resistance in all the compositions. Even Al-Mn alloy is less corrosive, but pure Mn has high corrosive resistance. Jutatip Namahoot et al. [24] expressed that low-volume fraction of intermetallic phases or high content of Mn in the matrix and the intermetallic phases can improve the corrosion resistance of Al-Mn alloys. This explains why Al-20%Cu-20-15%Mn is more prone than Al-22.5%Mn alloy composition. A high amount of pure metal phases was detected in Al-20%Cu-20–15%Mn composite as per the EDS analysis (Table 5), and this composition is a multi-alloyed composite with numerous phases and intermetallic.

## 4. Conclusions

Using microwave sintering and conventional sintering, three alloys (77.5Al-22.5Mn, 57.28Al-18.6Cu-24.12Mn, and 65Al-20Cu-15Mn) were sintered at 550 °C for 60 min. After sintering, they were aged at 150 °C, and corrosion tests were also under taken with the aged alloys. The following conclusion was made from this study.

Microwave sintered compacts exhibited higher densification factors and hardness when compared to conventional sintering. Microwave sintered alloys have finer microstructures and narrower pore sizes than conventional alloys, as evidenced by SEM microstructures. The results of EDS proved that the alloy formation of Mn with Al was more likely and was more prominent in the case of microwave sintering. Al-20%Cu-15%Mn showed less hardness and smaller grain size in the microwave sintering than Al-18.6%Cu-24.12%Mn. Higher corrosion resistance was observed in the case of microwave sintering due to the uniform dissolution of elements. Microwave sintering can be considered the best alternative to the conventional sintering technique for Al alloys processed via powder metallurgy.

## Figures and Tables

**Figure 1 molecules-26-03675-f001:**
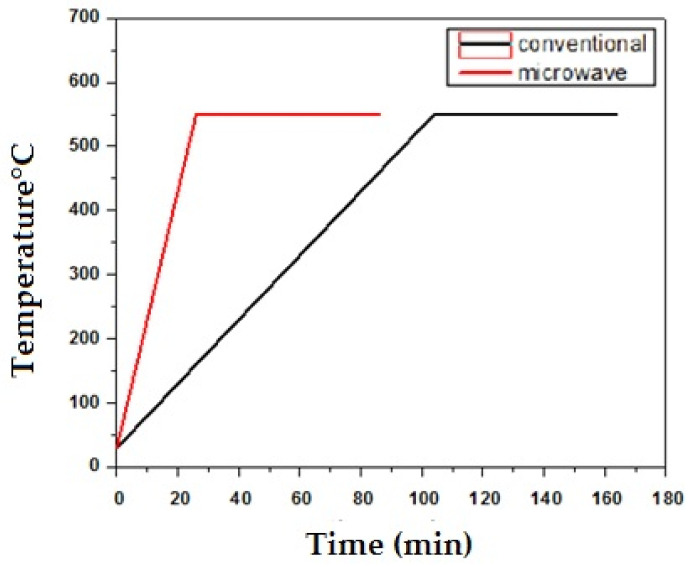
Heating profile of conventional sintering and microwave sintering.

**Figure 2 molecules-26-03675-f002:**
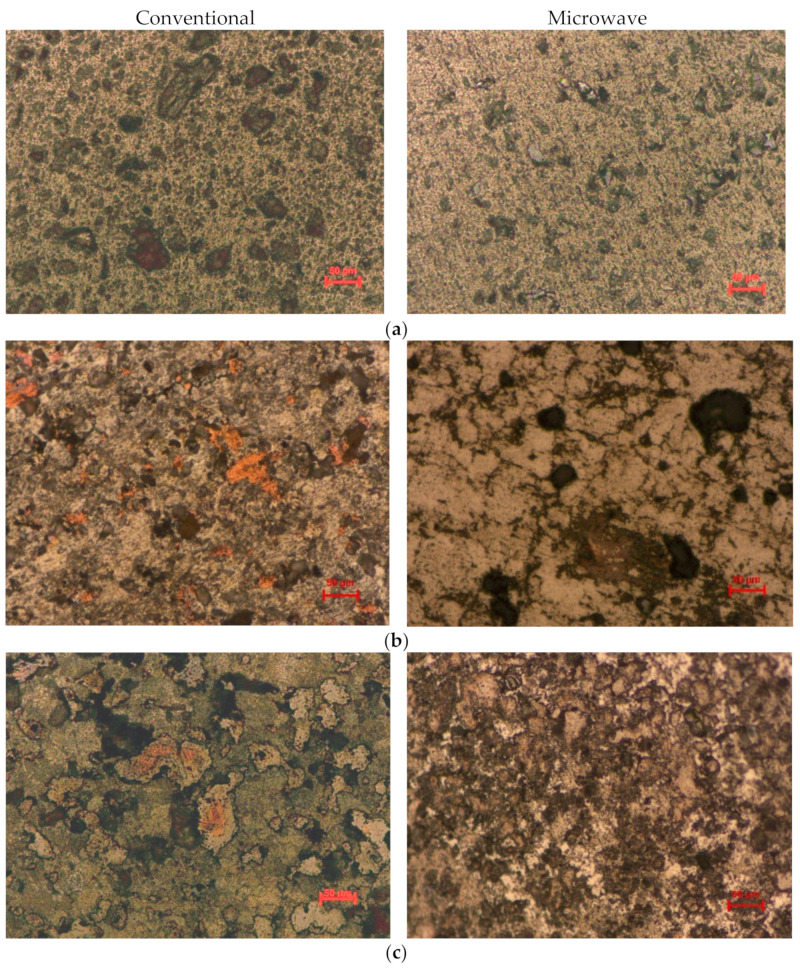
Optical microstructure of etched aluminum compacts of (**a**) Al-22.5%Mn; (**b**) Al-18.6%Cu-24.12%Mn; (**c**) Al-20%Cu-15%Mn sintered in conventional furnace (left) and microwave furnace (right).

**Figure 3 molecules-26-03675-f003:**
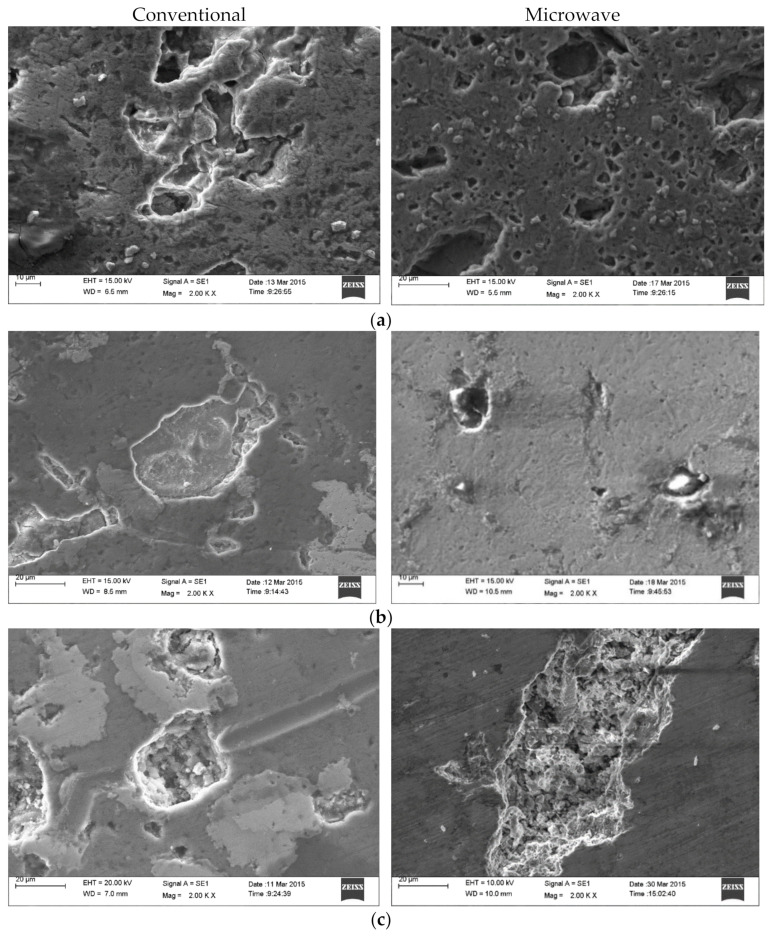
SEM microstructure of Aluminum compacts of (**a**) Al-22.5%Mn; (**b**) Al-18.6%Cu-24.12%Mn; (**c**) Al-20%Cu-15%Mn sintered in conventional furnace (left) and microwave furnace (right).

**Figure 4 molecules-26-03675-f004:**
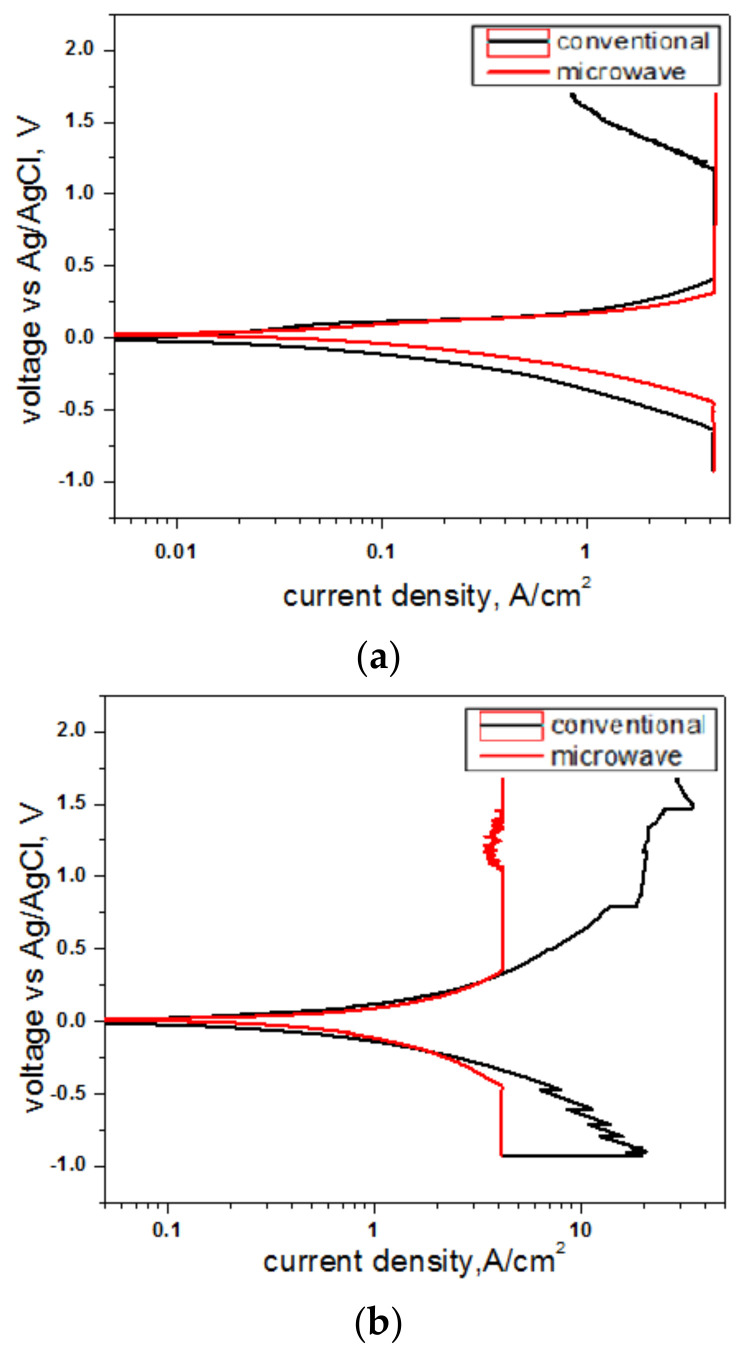
Comparison of the potentiodynamic polarization curves of conventionally and Microwave sintered (**a**) Al-22.5%Mn, (**b**) Al-18.6%Cu-24.12%Mn, and (**c**) Al-20%Cu-15%Mn composites.

**Table 1 molecules-26-03675-t001:** Powder Compositions concerning wt% used in the present study.

Sample No.	Aluminum Wt% *	Copper Wt% *	Manganese Wt% *
1	77.5	0	22.5
2	57.28	18.6	24.12
3	65	20	15

* Supplier: Krish Met Tech PVT. LTD, Chennai, India.

**Table 2 molecules-26-03675-t002:** Cumulative powder size of Aluminum.

Cumulative Powder Size	μm
D10	12.6
D50	31.0
D90	44.2

**Table 3 molecules-26-03675-t003:** Effect of compaction pressure and sintering on the densification response of Al-alloys compacted at 400 MPa and sintered at 550 °C.

Composition	Sintering Mode	Radial Shrinkage	Axial Shrinkage	% wt Loss	Green Density (% Theoretical)	Sinter Density (% Theoretical)	Densification Parameter
77.5Al-22.5Mn	Conventional	0.001	0.533	0.072	78.8	80.77	0.029
	Microwave	0.059	2.967	0.297	80.19	81.54	0.125
57.28Al-18.6Cu-24.12Mn	Conventional	0.039	0.125	0.34	71.12	71.51	0.013
	Microwave	0.078	0.152	3.721	70.4	73.30	0.096
65Al-20Cu-15Mn	Conventional	0.023	0.035	0.097	71.94	72.06	0.004
	Microwave	0.551	4.189	0.194	71.4	75.40	0.138

**Table 4 molecules-26-03675-t004:** Effect of heating mode on Vickers hardness of Al alloy compacts at a 50 gm load for 10 s dwell time.

Composition	Conventional	Microwave
77.5Al-22.5Mn	52.67 ± 5 Hv	86.65 ± 5 Hv
57.28Al-18.6Cu-24.12Mn	94.46 ± 10 Hv	745.8 ± 10 Hv
65Al-20Cu-15Mn	114.03 ± 5 Hv	580.37 ± 10 Hv

**Table 5 molecules-26-03675-t005:** Weight percentages of elements resulted from EDS analysis of particulate compacts of Al-alloys sintered in conventional and microwave furnaces.

Sample	Al-22.5%Mn		Al-18.6%Cu-24.12%Mn		Al-20%Cu-15%Mn	
	Conventional	Microwave	Conventional	Microwave	Conventional	Microwave
Al wt %	97.65	98.16	81.98	54.70	39.52	60.57
Cu wt %	0	0	1.79	22.73	13.99	20.78
Mn wt %	2.35	1.84	16.22	22.56	46.48	18.63

**Table 6 molecules-26-03675-t006:** Passivity parameters obtained for sintered compacts of Al-alloys of Cu and Mn from the anodic polarization study in 0.1N HCl. All the compositions were sintered at 550 °C in a conventional as well as microwave furnace.

Composition	Sintering Mode	Icorr	Ecorr	Corrosion Rate
		(mA/cm^2^)	(V)	(mpy)
77.5Al-22.5Mn	Conventional	1.135	0.003	1.407
	Microwave	1.103	−0.034	1.355
57.28Al-18.6Cu-24.12Mn	Conventional	2.171	0.005	2.398
	Microwave	1.672	−0.14	1.803
65Al-20Cu-15Mn	Conventional	1.29	0.019	1.276
	Microwave	1.164	−0.002	1.101

## Data Availability

Data available on request due to restrictions. The data presented in this study are available on request from the corresponding author.

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
