# Peer review of "Effect of Microwave and Conventional Modes of Heating on Sintering Behavior, Microstructural Evolution and Mechanical Properties of Al-Cu-Mn Alloys"

_molecules, 2021, doi:10.3390/molecules26123675_

Round 1

Reviewer 1 Report

Effect of microwave and conventional modes of heating on sintering behavior, microstructural evolution and mechanical properties of al-cu-mn alloys is analyzed in this paper. This work is good and the writing is also good. There are only some minor problems that suggested the author to modify, which are as follows:

  1. Abstract. The word “Microwave”. The first letters of this word may not need to be written in capital.
  2. There should be a space between numbers and units. The authors should check line 44, 87, 88, 90, 91, 113, 128, 130, 132, etc.
  3. The resolution of the formula 4 was too low.
  4. Although the quality of the products obtained by the microwave treatment were higher than that of the products obtained by the conventional treatment, the author could give the processing time and energy consumption of the two methods and make a comparison of them, to make the readers have a better understanding of the advantages and disadvantages of these two methods.

Author Response

Response to reviewer 1 comments:

Effect of microwave and conventional modes of heating on sintering behavior, microstructural evolution and mechanical properties of al-cu-mn alloys is analyzed in this paper. This work is good and the writing is also good. There are only some minor problems that suggested the author to modify, which are as follows:

1. Abstract. The word “Microwave”. The first letters of this word may not need to be written in capital.

Response: The authors have modified the word as ‘microwave’ in the revised manuscript.

2. There should be a space between numbers and units. The authors should check line 44, 87, 88, 90, 91, 113, 128, 130, 132, etc.

Response: As suggested by the reviewer, the authors have included spaces between the number and units

3. The resolution of the formula 4 was too low.

Response: The resolution has been improved

4. Although the quality of the products obtained by the microwave treatment were higher than that of the products obtained by the conventional treatment, the author could give the processing time and energy consumption of the two methods and make a comparison of them, to make the readers have a better understanding of the advantages and disadvantages of these two methods.

Response: The heating profiles of the compacts were as shown in Figure 1. The overall heating rate obtained in the microwave furnace was 30°C/minutes at 550°C for 60 minutes. Regarding the heating cycle, a 34% reduction in the processing time was obtained during microwave sintering against the slower heating rate (5°C/min) of a conventional furnace.

Reviewer 2 Report

It is an interesting paper about the benefits of sintering using microwaves. In my opinion, some minor changes should be addressed:

The youngs modulus.

Hv formula cannot be seen properly

Parameters and formulas should be in cursive

Probably Table 3 should be commented previously to Table 4

Line 176 of the cu help

Review English generally f.i. more minor line 219

Author Response

Response to reviewer 2 comments:

It is an interesting paper about the benefits of sintering using microwaves. In my opinion, some minor changes should be addressed:

The young’s modulus.

Response: The authors have corrected the sentence in the revised manuscript.

Hv formula cannot be seen properly

Response: We have replaced with math eq and the formula can be seen in the revised manuscript

Parameters and formulas should be in cursive

Response: All the formulas are modified in the revised manuscript.

Probably Table 3 should be commented previously to Table 4

Response: Table 3 is addressed before Table 4 in densification section as suggested by the reviewer

Line 176 of the cu help

Response: The authors have modified the line in the revised manuscript.

Review English generally f.i. more minor line 219

Response: Necessary changes have been included in the revised manuscript